# Target Selection Strategies for Demucs-Based Speech Enhancement

Caleb Rascon* and Gibran Fuentes-Pineda

Department of Computer Science, Instituto de Investigaciones en Matemáticas Aplicadas y en Sistemas, Universidad Nacional Autónoma de México, Coyoacán 04510, Mexico; gibranfp@unam.mx
* Correspondence: caleb@unam.mx

**Featured Application: Online audio applications such as real-time automatic speech recognition, sound source localization in service robotics, hearing prosthesis, mobile telecommunication, and teleconferencing.**

**Abstract:** The Demucs-Denoiser model has been recently shown to achieve a high level of performance for online speech enhancement, but assumes that only one speech source is present in the fed mixture. In real-life multiple-speech-source scenarios, it is not certain which speech source will be enhanced. To correct this issue, two target selection strategies for the Demucs-Denoiser model are proposed and evaluated: (1) an embedding-based strategy, using a codified sample of the target speech, and (2) a location-based strategy, using a beamforming-based prefilter to select the target that is in front of a two-microphone array. In this work, it is shown that while both strategies improve the performance of the Demucs-Denoiser model when one or more speech interferences are present, they both have their pros and cons. Specifically, the beamforming-based strategy achieves overall a better performance (increasing the output SIR between 5 and 10 dB) compared to the embedding-based strategy (which only increases the output SIR by 2 dB and only in low-input-SIR scenarios). However, the beamforming-based strategy is sensitive against the location variation of the target speech source (decreasing the output SIR by 10 dB if the target speech source is located only 0.1 m from its expected position), which the embedding-based strategy does not suffers from.

**Keywords:** Demucs; target selection; embeddings; phase-based beamforming

## 1. Introduction

When capturing speech in real-life scenarios, other audio sources are expected to also be present in the acoustic scene, resulting in the capture of a mixture of both target source and these other audio sources (also known as interferences and noise). It is then of interest to separate the speech source of interest by removing these other audio sources from the captured mixture using "speech enhancement" techniques, which recently have been carried out by means of deep learning with impressive results [1]. When carried out offline (i.e., by the use of previously captured audio recordings), several areas of applications have benefited, such as security [2] and music recording [3,4]. Furthermore, it is also of interest to carry out speech enhancement in an online manner (i.e., by the use of live audio capture). It is important to clarify that we differentiate between running a speech enhancement technique in an online manner and running it in "real-time". While the former only explicitly requires that the computation time is less than the capture time, the latter also employs additional application-dependent latency or response time requirements [5,6]. There are a varied set of applications that are to benefit from online speech enhancement, such as real-time automatic speech recognition [7], sound source localization in service robotics [8], hearing prosthesis [9], mobile telecommunication [10], and teleconferencing [11]. It would be difficult to impose additional response time requirements in a general sense to all these

applications. Thus, we chose to employ only the computation-time-less-than-capture-time requirement for this work.

The Demucs-Denoiser model [12] is an online speech enhancement technique that has been recently shown to outperform various other state-of-the-art techniques [13]. It not only provided a good enhancement performance (output signal-to-interference ratio above 20 dB in most cases), but was able to do so with very small window segments (64 ms). This makes it a very good representative of the current state of the art in online speech enhancement.

However, in the same manner as any other speech enhancement technique, the Demucs-Denoiser model bears an important limiting assumption: that there is only one speech source present in the mix. In applications such as mobile telecommunication [10] and some teleconferencing scenarios [11], this may not be an such an issue given their one-user-at-a-time nature. Nevertheless, in applications such as service robotics [8] and hearing prosthesis [9], where more than one speech source is expected to be present, the use of speech enhancement techniques may not be appropriate. If more than one speech source is present, another set of techniques, known as "speech separation" [14], could be applied, since they aim to separate the various speech sources present in the mix into their own separate audio stream. However, speech separation techniques are not known to be able to be run in an online manner while speech enhancement techniques can [13].

It has been shown that when feeding a mixture of speech sources to current online speech enhancement techniques, some separate the speech mixture from the rest of the nonspeech sources, while others (like the Demucs-Denoiser model) seem to separate the "loudest" speech source [13]. It is of interest, then, to carry out target selection as part of the speech enhancement process, such that the speech source of interest is the one being enhanced, while considering the rest of the speech sources as interferences.

To this effect, two target selection strategies are proposed in this work:

1. Embedding-based: A type of voice identity signature is computed from a previously captured audio recording of the target speech source. This, in turn, is used as part of the Demucs-Denoiser model to "nudge" its enhancement efforts towards the target speech source.
2. Location-based: The target speech source is assumed to be located in front of a two-microphone array. This two-channel signal is then fed to a spatial filter (also known as a beamformer) that aims to amplify the target speech source presence in the mixture to then be enhanced by the Demucs-Denoiser model.

This work is structured as follows: Section 2 describes the original Demucs-Denoiser model; Section 3 details the proposed target selection strategies; Section 4 describes the training of the Demucs-Denoiser model while using each of the target selection strategies; Section 5 details the evaluation methodology for both strategies; Section 6 presents the obtained results; Section 7 provides insights that were obtained from the results; and Section 8 concludes with potential future work.

## 2. The Demucs-Denoiser Model

The Demucs-Denoiser model [12] carries out speech enhancement in the waveform domain, and is based on a U-Net architecture [15], summarized in Figure 1.

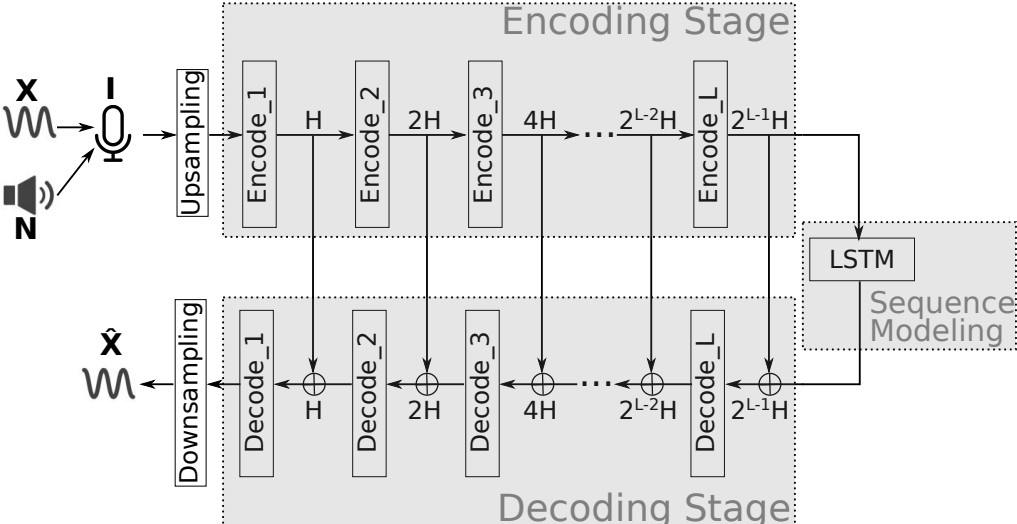

**Figure 1.** The Demucs-Denoiser model.

It is composed of three stages:

1.  **Encoding stage.** This stage consists of a series of $L$ meta-layers, referred to as "encoding layers", which are described in Figure 2. As can be seen, each encoding layer has (a) a 1-dimensional convolutional layer with a kernel size $K$, stride $S$, and $2^{l-1}H$ output channels (where $K$, $S$, and $H$ are all tunable parameters, and $l$ is the encoding layer index); (b) a rectified linear unit (ReLU) layer [16]; (c) another 1-dimensional convolutional layer, with a kernel size of 1, stride of 1, and $2^{l}H$ output channels; and (d) a gated recurrent unit (GRU) layer [17] whose objective is to force it to have $2^{l-1}H$ output channels (the same as its first convolutional layer). The output of each meta-layer is fed to both the next encoding layer (or, if it is the last one, to the sequence modeling stage), as well as to its corresponding "decoding layer" in the decoding stage of the model. The latter is referred to as a "skip connection".

2.  **Sequence modeling stage.** The output of the last encoding layer (meaning of the encoding stage in general) is a latent representation $z_I$ of the model's input $I$ (where $I = X + N$, where $X$ is the source of interest and $N$ is the noise to be removed). The $z_I$ latent representation is fed to this stage, the objective of which is to predict the latent representation $z_{\hat{X}}$ of the enhanced signal $\hat{X}$. Thus, it is crucial that the dimensions of its output are the same as of its input, so that the decoding stage is applied appropriately. In the causal version of the Demucs-Denoiser model (which is the one of interest for this work), this stage contains a stack of $B$ number of long short-term memory (LSTM) layers, with $2^{L-1}H$ number of hidden units, meaning that the number of hidden units (considered as the output of the sequence modeling stage) is equivalent to the number of output channels of the last encoding layer (i.e., the output of the decoding stage in general).

3.  **Decoding stage.** The objective of this stage is to decode the latent representation $z_{\hat{X}}$ (created by the sequence modeling stage) back to the waveform domain, resulting in the enhanced signal $\hat{X}$. This stage comprises another series of $L$ meta-layers, referred to as "decoding layers", which are described in Figure 3. As can be seen, each decoding layer carries out the reverse of its corresponding encoding layer, being composed of (a) a 1-dimensional convolutional layer with a kernel size of 1, stride of 1, and $2^{l}H$ output channels, (b) a GRU layer that converts the number of output channels to $2^{l-1}H$, (c) a 1-dimensional transposed convolutional layer with a kernel size $K$, stride $S$, and $2^{l-2}H$ output channels, and (d) an ReLU layer. Its input is the element-wise addition of the skip connection from the corresponding $l$th encoding layer and of the output of the last decoding layer (or the output of the sequence modeling stage, if it is the first one). Additionally, the last decoding layer does not employ an ReLU layer

since its corresponding convolutional layer has only one output channel and, thus, is the output of the Demucs-Denoiser model ($\hat{X}$).

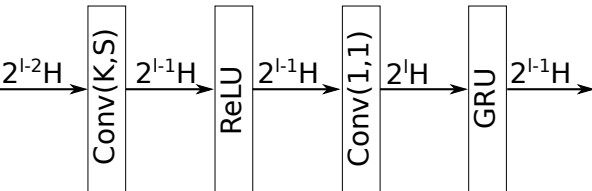

**Figure 2.** An encoding layer in the Demucs-Denoiser model.

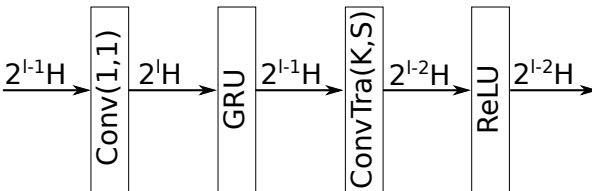

**Figure 3.** A decoding layer in the Demucs-Denoiser model.

The authors of [12] found that upsampling the input signal $I$ (and, respectively, downsampling the output signal $\hat{X}$ to $I$'s original sampling rate) improved the model's accuracy. This is carried out in an online manner by the use of a sinc interpolation filter [18].

This model provided very good enhancing results (output signal-to-interference ratios above 20 dB) with very small window segments (64 ms) [13], outperforming other online speech enhancement techniques such as spectral feature mapping using the mimic loss function [19] and MetricGAN+ [20]. This is evidence that the Demucs-Denoiser model is a prime representative of the current state of the art in online speech enhancement.

However, in the same evaluation [13], it was shown that all the evaluated techniques, including Demucs-Denoiser, suffered from considerable enhancement degradation when there was more than one speech source present in the input signal. Output signal-to-interference ratios dropped between 10 and 20 dB in the majority of the evaluated situations. To be fair, this is not surprising, since speech enhancement techniques assume only one speech source to be present [1] and, empirically, it seems that Demucs-Denoiser tends to enhance the loudest speech source [13], which may not always be the target speech. Thus, it is of interest to improve the enhancing performance of Demucs-Denoiser when there is more than one speech source present. This can be carried out by the use of target selection strategies, as described in the following section.

## 3. Proposed Target Selection Strategies

In this section, the two proposed target selection strategies for the Demucs-Denoiser model are described. The implementation of both target selection strategies can be publicly accessed at https://github.com/balkce/demucstargetsel (accessed on 12 June 2023).

### 3.1. Embedding-Based Target Selection

The embedding-based target selection strategy is summarized in Figure 4. As can now be seen, $I = X + N + Y$, where $X$ is still the source of interest and $N$ is still the noise to be removed, but now another speech source $Y$ is present.

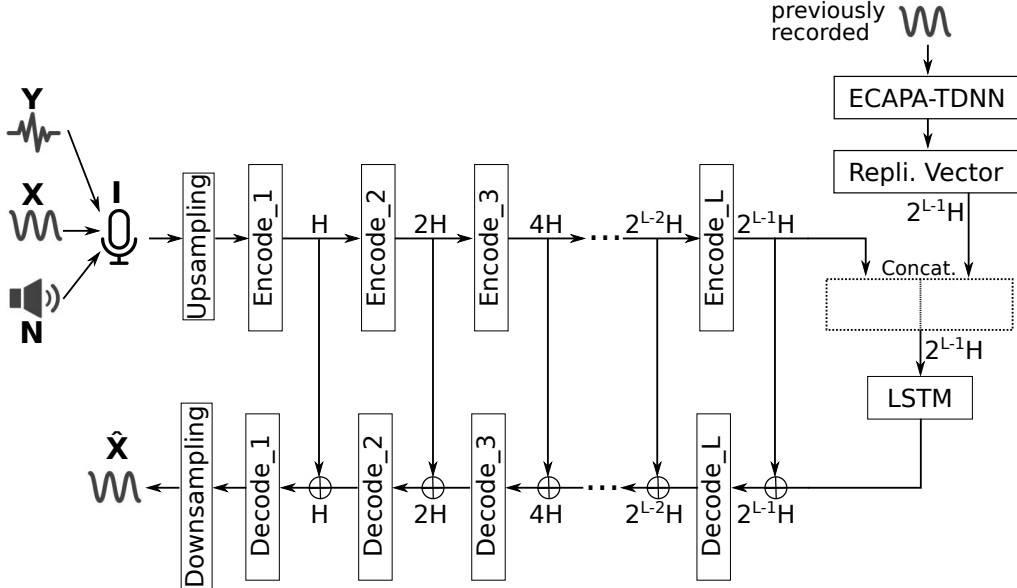

**Figure 4.** Embedding-based target selection strategy.

It is assumed that a previous recording of the target speech is known, so that an embedding of their speech can be extracted a priori. This embedding is then concatenated channel-wise to the input of the sequence modeling stage. However, the dimensions of the output of the sequence modeling stage are kept at $2^{L-1}H$, so as to not differ from its original operation. The objective of this concatenation is to "nudge" the focus of the sequence modeling towards the target speech, instead of the loudest speech source.

The embedding codifier chosen for this strategy is ECAPA-TDNN [21], which is based on the x-vector architecture [22,23]. The latter uses a time-delay neural network [24] trained for a speaker classification task, and employs the outputs of the "bottleneck" layer (the one preceding the final fully-connected layer) as the codified embedding. The ECAPA-TDNN technique further extends this idea by:

- Pooling channel- and context-dependent statistics to be used as part of its temporal attention mechanism, to focus on speaker characteristics that occur in different time segments.
- Creating a global context by joining local features with the input's global properties (mean and standard deviation), so that the attention mechanism is more robust against noise.
- Rescaling features from each time segment to benefit from the previously described global context.
- Aggregating all the aforementioned features, as well as the ones from the first layers of the TDNN, and achieving this by way of summation into a static embedding size (in this case, 192 dimensions), regardless of the input size.

The ECAPA-TDNN technique has been successfully employed, not only for speaker verification tasks [25,26], but also for speech synthesis [27], for depression detection in speech [28], and speech stuttering detection [29]. This indicates that ECAPA-TDNN is quite versatile in its ability to encode the speaker identity in an embedding.

It is important to mention that this type of target selection scheme is not new. The works of [30,31], as well as of [32], carry out a similar strategy. However, their underlying speech enhancement techniques rely on time–frequency masking, while the Demucs-Denoiser model used in this work completely relies on time domain. In [13] it was shown that this provides low response times and better performance against reverberation compared to other time–frequency-masking speech enhancement techniques.

### 3.2. Location-Based Target Selection

The location-based target selection strategy is summarized in Figure 5. As can be seen, two input signals ($I_1$ and $I_2$) are captured from a two-microphone array. And in this case, $I_1 = X + N_{\Phi_1} + Y_{\Phi_1}$ and $I_2 = X + N_{\Phi_2} + Y_{\Phi_2}$, where $X$ is still the source of interest, $N$ is still the noise to be removed, and $Y$ is still another speech source, but both $N$ and $Y$ arrive at different moments in time (represented by $\Phi_1$ and $\Phi_2$, respectively) at each microphone.

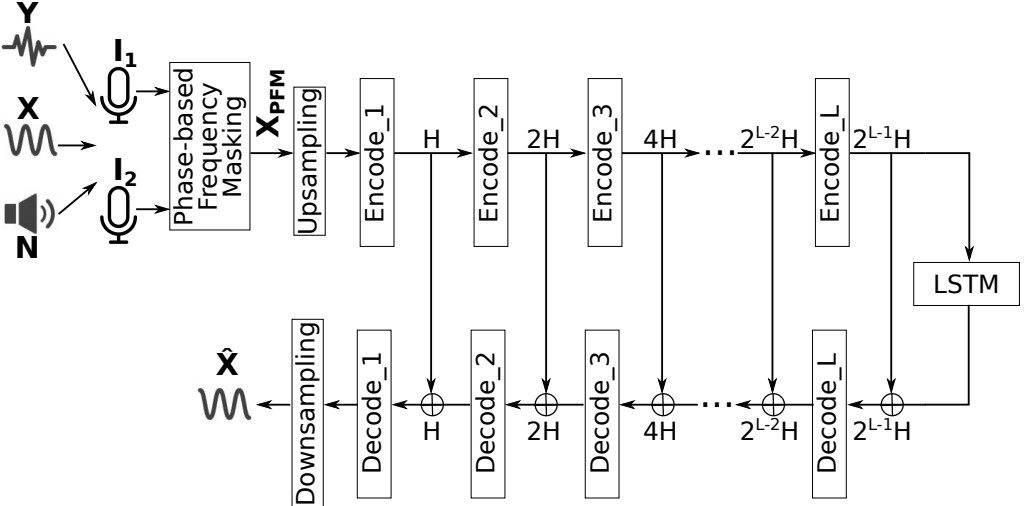

**Figure 5.** Location-based target selection strategy.

It is assumed that the target speech source is positioned in front of the two-microphone array. A lightweight spatial filter (also known as a beamformer) steered towards that direction is used to amplify the presence of the target speech such that it is the loudest speech source in the output mixture. This is then fed to the Demucs-Denoiser model which has been shown to enhance the loudest speech source in a given mixture [13]. Thus, thanks to the amplification carried out by the beamformer, the result is that the the target speech source is enhanced.

There are several beamforming techniques that can be used for this purpose, such as minimum variance distortionless response (MVDR) [33,34] or generalized sidelobe canceller (GSC) [35,36]. However, it has been shown [37] that a simple phase-based frequency masking (PFM) [38] is able to obtain a good amplification of the speech source at a given location while still being able to run in an online manner.

Like any other beamformer, PFM relies on knowing a prori in what direction the target source is located relative to the microphone array. This is to calculate a series of phase shift weights that are applied to the input signals such that any information coming from the target direction is aligned and, thus, can be amplified by different means. In the case described in this work, such weights are not required to be calculated since the direction of the target source is assumed to be in front of the microphone array. Thus, the input signals are already aligned towards the target direction. Following this, a frequency binary mask ($\Omega$) is calculated by employing Equation (1).

$$\Omega[f] = \begin{cases} 1, & \text{if } |\phi(I_1[f]) - \phi(I_2[f])| < \sigma \\ 0, & \text{otherwise} \end{cases} \tag{1}$$

where $f$ is the frequency index (of $F$ total frequency bins), $I_1$ and $I_2$ are the Fourier transform of the first and second input signals (respectively), the $\phi(\cdot)$ function calculates the phase from a given frequency component, and $|\cdot|$ is the absolute value mathematical operator. Finally, $\sigma$ is the phase difference threshold, such that if the phase difference at frequency $f$ is below it, it means that the frequency component at $f$ belongs to the target source; otherwise, it belongs to an interference.

The beamformer output $X_{PFM}$ is calculated using Equation (2).

$$X_{PFM}[f] = I_1[f]\Omega[f] \tag{2}$$

It is important to mention that the $\sigma$ parameter needs to be tuned. Small values of $\sigma$ may provide a better isolation of the target source, but this occurs at the expense of increased sensitivity to variation in the target location. Values between 10° and 30° have been shown to provide a good balance between these two issues [37].

Additionally, since the resulting frequency mask is binary, this beamformer introduces a considerable amount of discontinuities in the frequency domain. This, in turn, results in a considerable amount of musical artifacts in its time-domain output, which need to be removed by the subsequent Demucs-Denoiser model.

This location-based target selection strategy is very similar to a previous work, described in [38], where the beamformer output as well as its counterpart (where the frequency mask is inverted to obtain an estimation of the interferences) were fed into a bidirectional LSTM network. The objective of this network was to use both outputs to carry out a type of interference cancellation. It provided good results; however, it was trained and evaluated only using simulated signals, while the Demucs-Denoiser model has been shown to work in real-life scenarios [12,13].

## 4. Training Methodology

The employed training methodology is based on the one employed to train the original Demucs-Denoiser model [12]. The clean and noise recordings from the Interspeech 2020 deep noise suppression challenge [39] were utilized to create 200 h of data to be used for training, validation, and evaluation. The original data creation scripts were heavily modified to include relevant information to train both of the previously described target selection strategies.

To simulate the reverberation and microphone array, we made use of the Pyroomacoustics package [40]. Several data creation parameter combinations (later described) were used to create a series of input data instances: a set of artificial recordings employed for training, validating, and evaluating both target selection strategies, representing the given parameter combination.

For each input data instance, a $4 \times 6$ m room was simulated with a two-microphone array positioned in the center of the room, and the following steps were carried out to create its artificial recordings:

1. Randomly select an inter-microphone distance between a given inter-microphone distance range of 0.05 and 0.21 m.
2. Randomly select a user $u$ from the set of clean recordings.
3. Randomly select a recording $r_u$ from user $u$.
4. Randomly select a 5 s segment from recording $r_u$ with an energy level above a given energy threshold of 0.6.
5. Establish the location $o_u$ of the source of $r_u$ 1 m away from the microphone array in the simulated room.
6. Randomly select a signal-to-noise ratio (SNR) between a given SNR range of 0 and 20 dB.
7. Randomly select a noise recording $r_n$.
8. Randomly select a 5 s segment from recording $r_n$.
9. Apply the SNR to the noise segment.
10. Randomly select a location $o_n$ of the source of $r_n$ in the simulated room.
11. Randomly select a number of interferences $E$, which are other users from the clean recordings, with a maximum of 2.
12. If $E > 0$:
    (a) Randomly select an input signal-to-interference ratio (SIR) between a given SIR range of 0 and 20 dB.

    (b)    Randomly select the users that will act as interferences $e$, which should be different from $u$.

    (c)    For each interference $e$:

        i.    Randomly select a recording $r_e$ from user $e$.

        ii.    Randomly select a 5 s segment from recording $r_e$ with an energy level above a given energy threshold of 0.6.

        iii.    Randomly select a location $o_e$ of the source of $r_e$ in the simulated room.

        iv.    Apply the SIR to the segment of interference $e$.

13. Randomly select a reverberation time $rt_{60}$ between a given reverberation time range of 0.1 and 3.0 s.

14. Apply Pyroomacoustics to simulate the environment, with all the given audio segments (clean, noise, and interferences), the result of which is a two-channel recording (one channel for each simulated microphone).

15. The simulated microphones are fed to the PFM filter to obtain $X_{PFM}$.

For each input data instance the following is stored:

- The clean recording $r_u$.
- The output of the PFM filter $X_{PFM}$.
- The first channel of the Pyroomacoustics output $I_1$.
- A text file with the following information:

    – The location of the noise recording $r_n$.

    – The location of all interference recordings $r_e$.

    – The location of a randomly selected recording $r_{u_2}$ of user $u$, such that $r_{u_2} \neq r_u$, to create the embedding of user $u$.

- All the relevant data creation parameters (SNR, $E$, input SIR, and $rt_{60}$) are stored as part of the file name of the $X_{PFM}$ recording.

$X_{PFM}$ is used to train the location-based target selection strategy. $I_1$ and $r_{u_2}$ are used to train the embedding-based target selection strategy. It is important to note that since $r_{u_2} \neq r_u$, the embedding is not created from $r_u$ (the clean recording of user $u$); this hopefully provides generality in its use.

Three datasets were created: 80% of the users for training, 10% of the users for validation (from which the best model is kept), and 10% of the users for evaluation (from which the evaluation metrics were calculated).

In terms of training parameters, the Adam optimizer [41] was used, with a learning rate of $3e^{-4}$, a $\beta_1$ of 0.9, and a $\beta_2$ of 0.999. The training was run for 75 epochs. If after 10 epochs, the validation loss did not improve, the learning rate was halved, to refine the optimization search space.

The employed loss function is the same as the original implementation, presented in [12]. It is the weighted sum of the $\ell_1$ metric between $X$ and $\hat{X}$, and a multiresolution short-time Fourier transform (STFT) loss function ($MuSTFT$), as presented in Equation (3).

$$loss = \ell_1(X, \hat{X}) + \gamma_{Mu} MuSTFT(X, \hat{X}) \tag{3}$$

where

$$\ell_1(X, \hat{X}) = \frac{\sum_f^F |X[f] - \hat{X}[f]|}{F} \tag{4}$$

and

$$MuSTFT(X, \hat{X}) = \gamma_{Sp} \frac{\sum_\theta^\Theta SpecConv(X, \hat{X}, \theta)}{\Theta} + \gamma_{Lo} \frac{\sum_\theta^\Theta LogMag(X, \hat{X}, \theta)}{\Theta} \tag{5}$$

which, in turn, gives

$$SpecConv(X, \hat{X}, \theta) = \frac{\sum_f^F \sum_t^T (X_\theta[t,f] - \hat{X}_\theta[t,f])^2}{\sum_f^F \sum_t^T X_\theta[t,f]^2} \tag{6}$$

$$LogMag(X, \hat{X}, \theta) = \frac{\sum_f^F \sum_t^T |\ln X_\theta[t,f] - \ln \hat{X}_\theta[t,f]|}{FT} \tag{7}$$

where $\gamma_{Mu}$, $\gamma_{Sp}$, and $\gamma_{Lo}$ are loss weights (with values of 0.3, 0.5, and 0.5, respectively, in this work); $\theta$ is an STFT resolution (defined by a given Fourier size, window hop, and window size); $X_\theta$ and $\hat{X}_\theta$ are, respectively, the STFT'ed $X$ and $\hat{X}$ sampled at resolution $\theta$; $\Theta$ is the number of calculated resolutions; and $t$ is the time index (of $T$ total time bins).

The following are the hyperparameters used as part of the architecture of the trained model: number of initial output channels ($H$) of 64, kernel size ($K$) of 8, stride ($S$) of 0.5 s, number of encoding/decoding layers ($L$) of 5, and number of LSTM layers ($B$) of 2.

For replicability, the authors have made public the implementation of the training methodology and dataset creation, which can be accessed at https://github.com/balkce/demucstargetsel (accessed on 12 June 2023).

## 5. Evaluation Methodology

The output signal-to-interference ratio was calculated (in decibels) for each evaluation instance, as presented in Equation (8).

$$SIR = \frac{||P(\hat{X}, X)||^2}{\sum_e^E ||P(\hat{X}, Y_e)||^2} \tag{8}$$

where $Y_e$ is the $e$th interference, and $P(mix, q)$ is a function that calculates the energy of $q$ inside the mixture. To calculate these energies, an optimization process is employed, popularly implemented in the *bss_eval_sources* package [42]. To perform this, unfortunately, it also requires all the $Y_e$ interferences to also be estimated. These estimations are not carried out as part of the speech enhancement process (which only estimates the source of interest). However, the *bss_eval_sources* also calculates the signal-to-distortion ratio, and it does so in such a manner that if it only is fed the clean signal $X$ and the estimated signal $\hat{X}$, this metric is equivalent to the output SIR presented in Equation (8). Thus, this is the method that was employed to calculate the metric of output SIR which, in turn, is used to measure enhancement performance.

*Distance Offset*

An additional evaluation dataset was also created to measure the robustness of the location-based target selection strategy against variations of the position of the source of interest.

The creation process of this additional dataset is similar to what is described in Section 4, with the additional parameter of varying the position of the source of interest distance from its expected position: 1 m away from the microphone array. This distance is referred to here as "distance from center". Ten hours of evaluation data were created, using 10% of the users intended for evaluation, with the following evaluation parameter value ranges:

- Number of interferences : [1, 2].
- Input SIR : 0 dB.
- Input SNR : 19 dB.
- Reverberation time : [0.1, 0.2] s.
- Horizontal distance from center: [−0.1, 0.1] m.
- Vertical distance from center: [−0.05, 0.05] m.
- Absolute distance from center: [0, 0.11] m (stored).

## 6. Results

As explained beforehand, the metric with which the enhancement performance is measured is the output signal-to-interference ratio (SIR). It is this performance metric that is observed and plotted against the other evaluation parameters (number of interferences, input SIR, etc.). These plots are in the form of error plots, in which, usually, the middle point indicates the median and the vertical line indicates a type of error against another metric. However, it is very important to note that, in the case of this work, the vertical lines indicate the standard deviation of the results. This was decided to provide the reader with a way to observe the result variability in each evaluation, and should only be considered for comparative purposes (not for any statistical analysis).

It is also important to note that each of the results shown here were obtained from a subset of the overall evaluation, to focus the discussion on what we believe is worthwhile presenting (and from which other parts of the evaluation may deviate). Thus, unless stated otherwise, these are the default values for each of the evaluation parameters when not being focused on or discussed:

- Number of interferences : 0.
- Input SIR :
  - Not applicable when there are no interferences.
  - Between 0 and 7 dB (inclusive), when there are interferences.
- Input SNR : greater than or equal to 15 dB.
- Reverberation time : less than or equal to 0.2 s.
- Distance from center : 0 m.

In the following figures and the rest of this writing, for simplicity, the embedding-based target selection strategy is referred to as "Demucs-Embed", the location-based target selection strategy is referred to as "Demucs-PhaseBeamform", and the original version of the Demucs-Denoiser model is referred to as just "Demucs".

### 6.1. Overall Output SIR

The overall output SIR is shown in Figure 6. The number of interferences is at 0 for this evaluation, with the objective to establish a baseline of only the speech enhancement side of both target selection strategies.

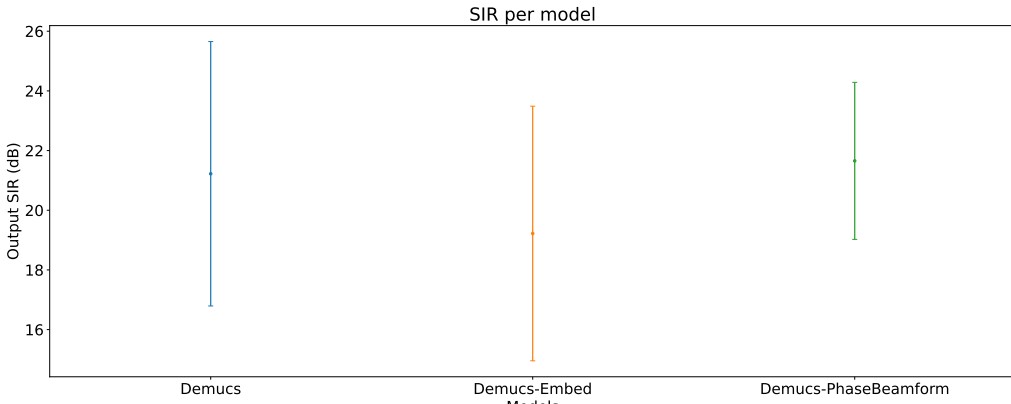

**Figure 6.** Overall output SIR.

As can be seen, Demucs-Embed is not able to perform its speech enhancement objective as well as the original Demucs. This may be because the sequence modeling stage of the Demucs-Denoiser model now needs to carry out a double duty (target selection and speech enhancement), resulting in refocusing resources that would have been spent carrying out speech enhancement to now carry out target selection. Thus, as can be observed in Section 6.2, the speech enhancement performance is diluted to provide a better target

selection. The reader may be interested to know that additional experiments were carried out with an increased number of LSTM layers, so as to provide more resources to the sequence modeling stage. However, such an increase did not improve upon the results shown in Figure 6. Additionally, it is noteworthy that the performance reduction is not considerable as the difference of the median values is around 2 dB; thus, Demucs-Embed is still providing (for the most part) a good enhancement performance.

With regard to Demucs-PhaseBeamform, it can be seen that it not only provides a similar average performance to the original Demucs, but improves upon it by reducing the result variability.

### 6.2. Output SIR vs. Number of Interferences

The impact of the number of interferences on the output SIR can be seen in Figure 7.

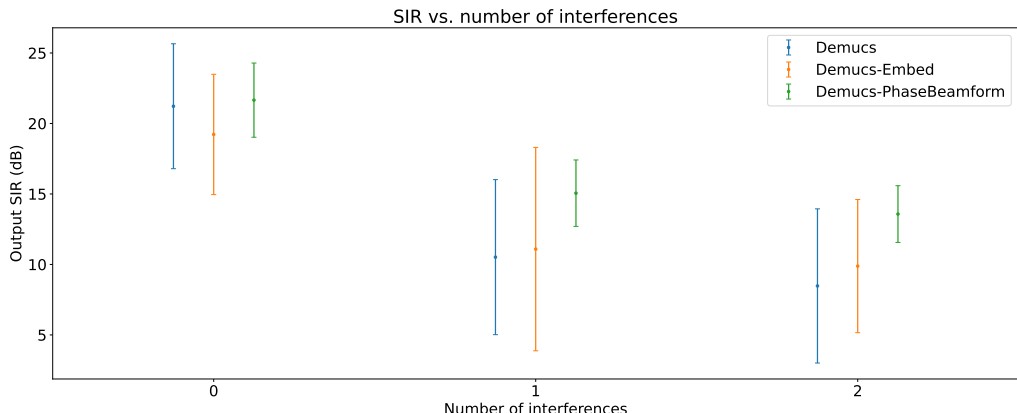

**Figure 7.** Output SIR vs. number of interferences.

For reference, Figure 6 is included in Figure 7 in the case of 0 interferences, and it can be observed that the overall enhancement performance is reduced considerably when interferences are present. However, both strategies were able to improve upon the average performance of the original Demucs, Demucs-PhaseBeamform providing the best performance in all cases in which there are interferences present.

### 6.3. Output SIR vs. Input SIR

The impact of the input SIR on the output SIR can be seen in Figure 8. In this evaluation, only one interference is present, to simplify the visualization of the results.

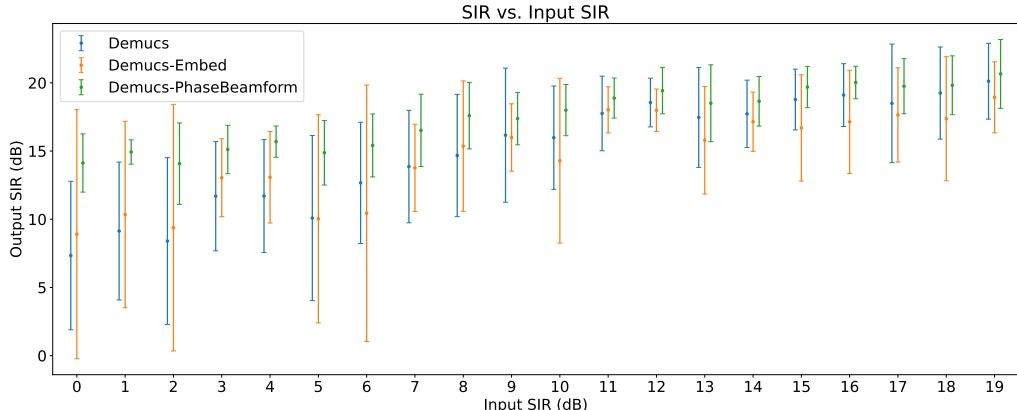

**Figure 8.** Output SIR vs. input SIR.

As can be seen, the greater the input SIR, the greater the output SIR. In the case of the Demucs-Embed, this target selection strategy is able to improve the average performance

of the original Demucs in the cases of very low input SIR (<5 dB), but it is not able to do so otherwise. This indicates that the target selection strategy is indeed able to "nudge", albeit slightly, the focus of the speech enhancement towards the source of interest. However, it is not able to outperform it when the source of interest is louder than the interferences (i.e., input SIR $\geq$ 5 dB), in which cases, the original Demucs is able to provide a better speech enhancement. Additionally, although the average performance is improved in low-input-SIR scenarios, the result variability does increase, which indicates some unpredictability when using Demucs-Embed.

With regard to Demucs-PhaseBeamform, it provides a better average performance and lower result variability compared to both Demucs and Demucs-Embed throughout all input SIR cases. Moreover, the positive relationship between output SIR and input SIR does not appear to be as strong, indicating that it is more robust against changes in input SIR. This may be because of the beamformer that precedes the Demucs-Denoiser model, which, in a way, is carrying out a type of input SIR normalization that provides this robustness.

### 6.4. Output SIR vs. Input SNR

The impact of the input SNR on the output SIR can be seen in Figure 9. No interferences were present for this evaluation.

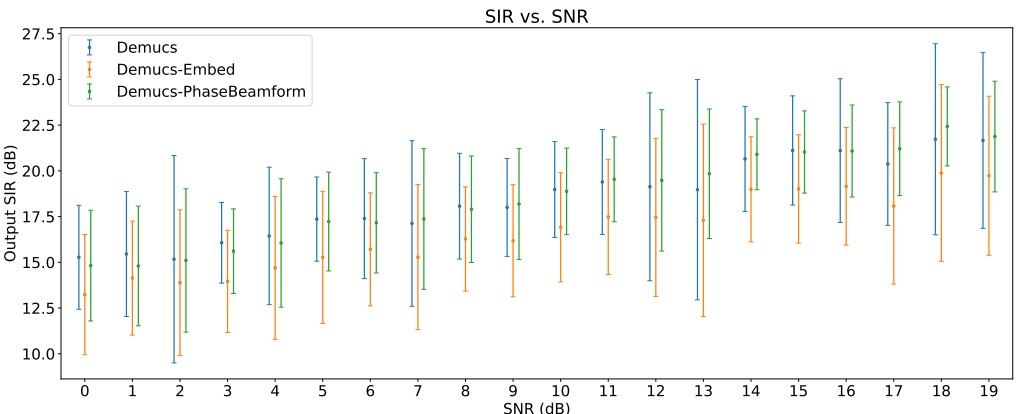

**Figure 9.** Output SIR vs. input SNR.

Furthering what was discussed in Section 6.1, it can be seen that the reduction of speech enhancement performance of Demucs-Embed compared to the original Demucs is general throughout all of the evaluated values of input SNR. As mentioned before, this may be caused by the refocusing of resources into target selection during the sequence modeling stage of the Demucs-Denoiser model, which results in enhancement performance dilution. Interestingly, the reduction in the average performance persists around 2 dB (as in Figure 6), which indicates that such reduction, although generalized, is not considerable.

With regard to Demucs-PhaseBeamform, its speech enhancement performance is closer to the original Demucs throughout all of the evaluated values of input SNR. However, in many cases, especially in moderate- to high-SNR scenarios (>10 dB), the result variability is reduced, providing a more predictable behavior. Again, the beamformer is providing some sort of SNR normalization to the Demucs-Denoiser model which results in this predictability.

### 6.5. Output SIR vs. Reverberation Time

The impact of the reverberation time on the output SIR can be seen in Figure 10. To avoid saturating the plots, the standard deviation vertical bar is shown only every 10 reverberation time values. Again, no interferences were present for this evaluation.

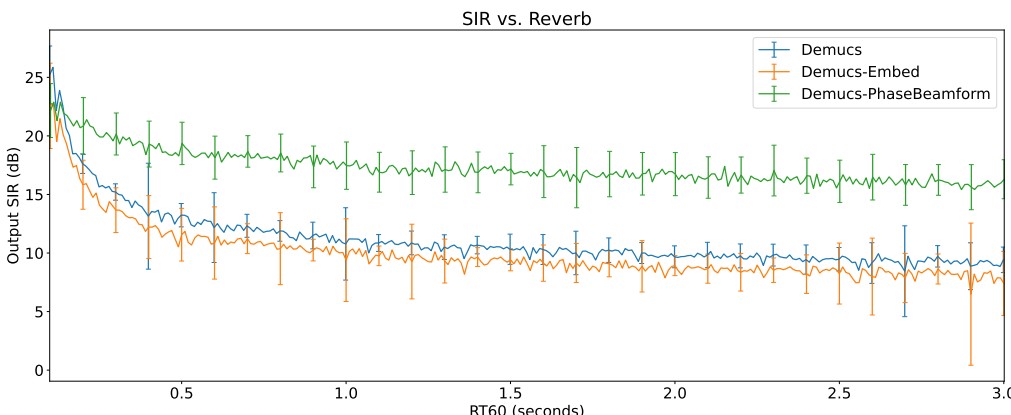

**Figure 10.** Output SIR vs. reverberation time.

The slight decrease of enhancement performance of Demucs-Embed compared to the original Demucs is now observed throughout all reverberation times, while Demucs-PhaseBeamform outperforms Demucs in every reverberation scenario. Since the only difference between them is the beamformer that precedes the Demucs-Denoiser model, it appears that this filter is complementing the model very well, resulting in reverberation robustness.

*6.6. Output SIR vs. Distance from Center*

The impact of the distance to center on the output SIR can be seen in Figure 11. Only Demucs-PhaseBeamform was evaluated, since Demucs-Embed employs the input of one of the microphones, not the output of the location-sensitive beamformer; thus, it should not be affected by this evaluation parameter. Furthermore, all of the 10-h dataset was used for this evaluation. As a reminder for the reader, this dataset employs the presence of interferences, along with low noise and small reverberation times. Thus, the results presented in Figure 11 are a type of general enhancement performance that could be used to compare in an overall manner the performance with no location variation and while varying the location of the source of interest.

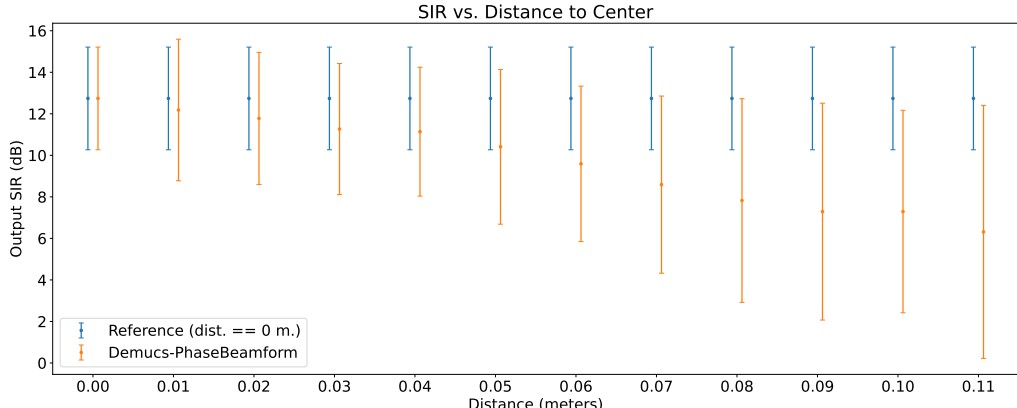

**Figure 11.** Output SIR vs. distance from center.

As can be seen, there is a considerable decrease of enhancement performance even with small distances (even as low as 0.05 m), with a considerable increase in result variability. This indicates that although the beamformer is able to complement the Demucs-Denoiser model well, it is only able to do so if the source of interest is located exactly where it is expected to be. Thus, the Demucs-Denoiser model, during training, appears to have inherited the sensitivity toward localization errors that the beamformer suffers from.

## 7. Discussion

As can be seen from the results shown in the previous section, Demucs-Embed is able to provide moderate improvements over the original Demucs (an increase of ∼2 dB output SIR); however, it does so in circumstances where the target speech and the interferences bear similar energy (<5 dB input SIR). In all other circumstances, the original Demucs consistently provides a better output SIR than when employing Demucs-Embed. Given that there is additional information that the Demucs-Denoiser model can use to select the target speech, an output SIR improvement over all circumstances was expected. Regarding why this is the case, the authors believe there are two possible reasons. First, the sequence modeling stage of the Demucs-Denoiser model is aimed specifically at speech enhancement, and it was designed as such using purely LSTM layers. This architecture may not be able to carry out both speech enhancement and target selection at the same time, which hints that other approaches or architectures for this stage should be explored in the future (some possible solutions are described in the following section). Second, it is important to remember that the original Demucs has been shown to perform well when one target speech source has a higher energy than the rest of the speech sources in the mix [13]. In these circumstances, no explicit target selection is actually carried out; thus, the sequence modeling stage of the original Demucs can focus solely on speech enhancement, while the sequence modeling stage of Demucs-Embed is still aiming to carry out both target selection and speech enhancement.

Additionally, it is important to remind the reader that Demucs-Embed assumes that there is a previously captured recording of the target speech source at hand, so as to calculate the required embedding for this strategy. This may not be viable in some circumstances, although, to be fair, it could be trivially obtained with a brief human–computer interaction prior to running the speech enhancement.

In any case, the results provide a strong incentive to only use Demucs-PhaseBeamform, since it not only provides a higher average performance compared to Demucs-Embed and the original Demucs, but also provides lower result variability. It should be noted that Demucs-PhaseBeamform also provides the potential to be used with a variety of microphone strategies, since it only requires a trivial reconfiguration of the beamformer to work. Only two microphones were considered here since, not only is it the lowest amount of microphones that could be used with this strategy (it can be expected that its performance will improve with a higher number of microphones), but currently a vast amount of teleconferencing and digital communication hardware employ a two-microphone array for their digital audio signal processing purposes. Indeed, the beamformer that precedes the Demucs-Denoiser model seems to be effectively supporting the Demucs-Denoiser model in several ways other than target selection: input SIR normalization, less result variability in high SNR scenarios, and even robustness against reverberation.

However, all the benefits that Demucs-PhaseBeamform offers come with an important caveat: the target speech source needs to be located very close to its expected position. Although the strategy seems to "tolerate" slight location deviations and still provide a good enhancement result (>10 dB in output SIR), distances from the expected position as small as 0.05 m result in important performance degradation. Several substrategies could be used to avoid this issue as future work, and are commented on in the following section.

## 8. Conclusions

Speech enhancement has been improved thanks in great part from deep-learning-based techniques and recently has grown in interest to be run in an online manner. The Demucs-Denoiser model has been shown to be a prime representative of the current state of the art of online speech enhancement, but is still limited by the assumption of having only one speech source present in the input mixture. In this work, two target selection strategies were proposed and evaluated: one that uses an embedding calculated from a previously captured recording of the target speech source (Demucs-Embed), and another that assumes

that the target speech source is located in front of a two-microphone array (Demucs-PhaseBeamform).

The results presented in this work point out that Demucs-Embed is effective in improving the original Demucs performance; however, it is not a considerable improvement ($\sim$ 2 dB output SIR improvement in the best of cases), and it can only be appreciated in low-input-SIR scenarios. This is quite possibly because the sequence modeling stage of the Demucs-Denoiser model is not designed to carry out both target selection and speech enhancement. Thus, other neural architectures could be explored, such as bidirectional LSTM layers or self-attention mechanisms. However, such exploration (proposed here as future work) should be carried out considering approaches that do not increase the response time considerably.

Demucs-PhaseBeamform consistently outperformed both Demucs-Embed and the original Demucs by a wide margin (by over 10 dB in low-input-SIR scenarios). Since other beamformers could be used, it would be of interest to explore their use as future work.

However, if the target speech source moves from where Demucs-PhaseBeamform assumes it is located, its performance degrades considerably ($>$ 5 dB drop in output SIR with a distance from the center of just 0.1 m). To avoid this issue, several substrategies are worth evaluating in future work: (a) train the Demucs-Denoiser model with data that were created with artificial location deviations so as to make it robust against them; (b) the chosen beamformer has a calibration parameter ($\sigma$, the phase difference threshold) that was established as static in this work and may provide higher tolerance by dynamically adjusting it via a given quality metric; and/or (c) incorporate a sound source localization mechanism as part of the strategy.

Additionally, no experiments were carried out in scenarios where interferences are located behind the target speech source. In these cases, both the interferences and the target speech source bear the same direction of arrival to the microphone array and, thus, will not be diminished by the beamformer. Hopefully, since the interferences tend to be behind the target speech source, they are located further away and, thus, may be quieter in the mix, which may point to their removal by the Demucs-Denoiser model. These issues are worth exploring as future work by the authors.

**Author Contributions:** Conceptualization, C.R. and G.F.-P.; methodology, C.R. and G.F.-P.; software, C.R.; validation, C.R.; formal analysis, C.R.; investigation, C.R.; resources, C.R. and G.F.-P.; data curation, C.R.; writing—original draft preparation, C.R.; writing—review and editing, C.R. and G.F.-P.; visualization, C.R.; supervision, C.R. and G.F.-P.; project administration, C.R.; funding acquisition, C.R. and G.F.-P. All authors have read and agreed to the published version of the manuscript.

**Funding:** This work was supported by PAPIIT-UNAM through the grant IA100222.

**Institutional Review Board Statement:** Not applicable.

**Informed Consent Statement:** Not applicable.

**Data Availability Statement:** The complete implementation of both strategies and trained weights, as well as dataset creation and evaluation methodologies, can be found at https://github.com/balkce/demucstargetsel (accessed on 12 June 2023).

**Acknowledgments:** The authors would like to acknowledge Alexandre Défossez, Gabriel Synnaeve, and Yossi Adi, the authors of the original Demucs-Denoiser model [12], for their efforts in making their implementation publicly accessible at https://github.com/facebookresearch/denoiser (accessed on 12 June 2023).

**Conflicts of Interest:** The authors declare no conflict of interest.

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
