# Peer review of "Target Selection Strategies for Demucs-Based Speech Enhancement"

_applsci, doi:10.3390/app13137820_

Round 1

Reviewer 1 Report

 Review

Target Selection Strategies for Demucs-Based Speech Enhancement

Caleb Rascon and Gibran Fuentes-Pineda

The manuscript proposes two methods for target selection strategies. The proposed methods are based on “Demucs-Model”. The authors offer two methods: “Demucs-Embed” and “Demucs-PhaseBeamform”. The first of them “Demucs-Embed”: voice identity signature is computed from a previously captured audio recording of the target speech source. It is used as part of the “Demucs-Model” to ‘nudge’ its enhancement efforts towards the target speech source. The second method Demucs-PhaseBeamform”: the target speech source is assumed to be located in front of a 2-microphone array. This 2-channel signal is processed by beamformer that aims to amplify the target speech source presence in the mixture to then be enhanced by the “Demucs-Model”. In the paper the results of comparative analysis of the two offered methods Demucs-Embed” and Demucs-PhaseBeamform” are presented. The paper consists of seven sections. The Section 1 is introduction. In the Section 2 the original “Demucs-Modelis described. In the Section 3 the proposed target selection methods are described. The Section 4 describes the training of the “Demucs-Model” while using each of the target selection strategies. In the Section 5 the evaluation methodology for both methods is presented. In the Section 6 the obtained results are analyzed. The Section 7 concludes by discussing the presented results.

I suppose that the presented results could be of interest for Journal Applied Sciences readers. In general, the manuscript is well-written, and I would support its publication after some major and minor improvements.

 Here are my few remarks.

 I) Major remark.

I was very surprised by the results presented in Figure 6, 7, 8, 9, 10. As follows from results presented in these figures the “Demucs-Embed” is not able to perform its speech enhancement objective as well as the original “Demucs”.

Why is it?

According to paper the method Demucs-Embed” include same algorithms as “Demucs” and signal preprocessing. The signal preprocessing is used to voice identity and enhancement efforts towards the target speech source.   

Where is “enhancement efforts towards the target speech source”?

There is no any improvement in the results presented in Figure 6-10. I do not understand this phenomenon. Authors added “voice identity” signal processing. The processing results become worse.

I mean that these results must be explained carefully and interpreted. I suppose that authors should to present the results with other parameters of “Training Methodology” when “Demucs-Embed” is better “Demucs”.

My point that there are situations when voice identity within framework of the Demucs-Embed” should to improve results of “Demucs”. Such situations should be presented in the paper.

 II) Minor remark.

 Line 41: “Demuc-Denoise” (must be “Demucs-Denoise”). I believe that it is better to use “Demucs” as in Section 6.  

 Line 47: “Demuc-Denoise” (must be “Demucs-Denoise”). I believe that it is better to use “Demucs” as in Section 6.  

 Line 82: three phases (it is better to use - three steps)

Word “phase” is used as signal parameter. The following words are more suitable: “stage“, “step”, “process”, “procedure”, etc.   

Line 83: Encoding phase (it is better to use - Encoding step)

Line 94: Sequence modeling phase (it is better to use - Sequence modeling step).

Line 106: Decoding phase (it is better to use - Decoding phase).

Line 109: ‘decoding layers” (must be decoding layers)

Line 143: Figure 4. Embedding-based target selection strategy.

The Figure 4 should be described carefully like Figure 1.

What is X, N, Y ?

Author should to point that I = X + N + Y etc.

Line 177: Figure 5. Location-based target selection strategy.

The Figure 5 should be described carefully like Figure 1.

What is X, N, Y ?

Author should to point that I1 = X + N + Y, I2 = X + N + Y, etc.

Line 424: Figure 10. Output SIR vs reverberation time.

There is no Legend 

Author Response

The authors greatly appreciate the comments from the reviewer. Here is a point-by-point response to the reviewer's comments:

- I was very surprised by the results presented in Figure 6, 7, 8, 9, 10. As follows from results presented in these figures the “Demucs-Embed” is not able to perform its speech enhancement objective as well as the original “Demucs”. Why is it? Where is “enhancement efforts towards the target speech source”? [...] My point that there are situations when voice identity within framework of the “DemucsEmbed” should improve results of “Demucs”. Such situations should be presented in the paper.
>>> Lines 458-472: It is important to note that these results also surprised the authors. More details were added about the reasons why the authors think that such results took place. In summary, the authors believe that the sequence modeling stage of the Demucs-Denoiser is not able to properly carry out both speech enhancement and target selection. Additionally, the Demucs-Denoiser model has shown that it is able to carry out good speech enhancement when the target speech source is louder than the rest of the speech sources (implicitely carrying out target selection in these circumstances). Thus, the Demucs-Denoiser model is able to outperform Demucs-Embed in such circumstances, since Demucs-Embed is trying to carry out both objectives, while Demucs-Denoiser model can solely focus on one.

- Line 41: “Demuc-Denoise” (must be “Demucs-Denoise”). I believe that it is better to use “Demucs” as in Section 6.
>>> Line 39: Typo fixed. The authors believe that using the term "Demucs-Denoiser" better describes the intent of the original authors, since the first "Demucs" model was used for musical instrument separation. In fact, their public github repository calls it "denoiser". Additionally, the authors want to differentiate between the Demucs-Denoiser model, and the two strategies that are complementing it. The authors believe that using the term "Demucs" to refer to the case with no strategies when comparing it to the other strategies is less confusing, while using "Demucs-Denoiser" to refer to the original methodology provides a simpler terminology for the reader to follow.

- Line 47: “Demuc-Denoise” (must be Demucs-Denoise). I believe that it is better to use “Demucs” as in Section 6.
>>> Line 46: Typo fixed.

- Line 82: three phases (it is better to use - three steps) Word “phase” is used as signal parameter. The following words are more suitable: “stage“, “step”, “process”, “procedure”, etc.
>>> Line 80: The word "stage" was chosen. Additionally, the word "phase" was replaced accordingly with "stage" throughout the manuscript.

- Line 83: Encoding phase (it is better to use - Encoding step)
>>> Line 81: The word "stage" was chosen.

- Line 94: Sequence modeling phase (it is better to use - Sequence modeling step).
>>> Line 92: The word "stage" was chosen.

- Line 106: Decoding phase (it is better to use - Decoding phase).
>>> Line 104: The word "stage" was chosen.

- Line 109: ‘decoding layers” (must be “decoding layers”)
>>> Line 107: Typo fixed.

- Line 143: Figure 4. Embedding-based target selection strategy. The Figure 4 should be described carefully like Figure 1. What is X, N, Y ? Author should to point that I = X + N + Y etc. 
>>> Lines 141-143: An explanation of X, N, and Y in Figure 4 was added.

- Line 177: Figure 5. Location-based target selection strategy. The Figure 5 should be described carefully like Figure 1. What is X, N, Y ? Author should to point that I1 = X + N + Y, I2 = X + N + Y, etc.
>>> Lines 177-181: An explanation of how I1 and I2 are conformed in Figure 5 was added.

- Line 424: Figure 10. Output SIR vs reverberation time. There is no Legend 
>>> Legend added to Figure 10.

Reviewer 2 Report

The authors of this manuscript propose to use the Demucs architecture to complement speech enhancement techniques with a target selection scheme, with the ultimate aim of ensuring that only the source of interest is enhanced.  The proposed methodology is based on two target selection strategies. The first is based on integration, using a coded sample of the target speech. The second strategy is localization-based, using a beamforming-based pre-filter to select the target that is in front of a two-microphone array. Although the results obtained show that the beamforming-based strategy performs better overall than the integration-based strategy, it is sensitive to variation in the location of the target speech source, which is not the case with the integration-based strategy.

The proposed manuscript is well structured and the written expression is generally fluent.
To definitively accept this submission, I propose the following minor revisions:

1. The abstract should be rewritten to get straight to the point. A sentence of context is sufficient. Next, the authors should explain the aims of the proposed work, describe its originality in relation to what already exists in the literature, and list their contributions. The most important point is to be quantitative in the results obtained, using representative metrics. On this last point, the authors need to do a thorough job.

2. In the general introduction, the literature review needs to be more in-depth:

2.1. How does your work compare with the following article https://doi.org/10.3390/s23094394?

2.2. The authors promote the virtues of supervised deep learning techniques. Why? Can unsupervised techniques be used in your case? Very interesting results have been obtained in biometrics in the following article https://doi.org/10.1109/JSEN.2021.3100151? Can the methodology used in this manuscript be adapted?

3. It is important to separate the discussion of the results obtained from the conclusions. These are two completely separate sections of the manuscript.

4. In the discussion, please specify the limitations of the methodology implemented in this manuscript.

5. The overall conclusion should be more quantitative, focusing on the relevant metrics obtained.

6. The written expression needs to be proofread by a native English speaker, as it is fraught with awkwardness. Today's writing includes turns of phrase that can be said orally.

The written expression needs to be proofread by a native English speaker, as it is fraught with awkwardness. Today's writing includes turns of phrase that can be said orally.

Author Response

The authors greatly appreciate the comments from the reviewer. Here is a point-by-point response to the reviewer's comments:

1. The abstract should be rewritten to get straight to the point. A sentence of context is sufficient. Next, the authors should explain the aims of the proposed work, describe its originality in relation to what already exists in the literature, and list their contributions. The most important point is to be quantitative in the results obtained, using representative metrics. On this last point, the authors need to do a thorough job.
>>> The abstract has been rewritten with the reviewers suggestions in mind.

2. In the general introduction, the literature review needs to be more in-depth:

2.1. How does your work compare with the following article https://doi.org/10.3390/s23094394?
>>> Lines 121-134: The work suggested by the reviewer is reference 13 of the manuscript which is cited extensively throughout the manuscript.

2.2. The authors promote the virtues of supervised deep learning techniques. Why? Can unsupervised techniques be used in your case? Very interesting results have been obtained in biometrics in the following article https://doi.org/10.1109/JSEN.2021.3100151? Can the methodology used in this manuscript be adapted?
>>> The reviewer's suggestion is appreciated by the authors, however, providing an in-depth literature review on the matter of unsupervised techniques used for speech enhancement is outside the scope of the manuscript.

3. It is important to separate the discussion of the results obtained from the conclusions. These are two completely separate sections of the manuscript.
>>> Lines 451-493: The results are now discussed in a separate section from the conclusions.

4. In the discussion, please specify the limitations of the methodology implemented in this manuscript.
>>> Lines 455-456, 488-492: The limitations of both target selection strategies are now specified in the discussion section.

5. The overall conclusion should be more quantitative, focusing on the relevant metrics obtained.
>>> Lines 505, 514, 517-518: Quantitative results are now presented in the conclusion section.

6. The written expression needs to be proofread by a native English speaker, as it is fraught with awkwardness. Today's writing includes turns of phrase that can be said orally.
>>> The manuscript was extensively revised and proofread.